# k-Nearest Neighbors by Means of Sequence to Sequence Deep Neural Networks and Memory Networks

## Abstract

k-Nearest Neighbors is one of the most fundamental but effective classification models. In this paper, we propose two families of models built on a sequence to sequence model and a memory network model to mimic the k-Nearest Neighbors model, which generate a sequence of labels, a sequence of out-of-sample feature vectors and a final label for classification, and thus they could also function as oversamplers. We also propose 'out-of-core' versions of our models which assume that only a small portion of data can be loaded into memory. Computational experiments show that our models outperform k-Nearest Neighbors, a feedforward neural network and a memory network, due to the fact that our models must produce additional output and not just the label. As an oversampler on imbalanced datasets, the sequence to sequence kNN model often outperforms Synthetic Minority Over-sampling Technique and Adaptive Synthetic Sampling.

## 1 Introduction

Recently, neural networks have been attracting a lot of attention among researchers in both academia and industry, due to their astounding performance in fields such as natural language processing Turian et al. (2010) Mikolov et al. (2013) and image recognition Krizhevsky et al. (2012)Deng et al. (2009). Interpretability of these models, however, has always been an issue since it is difficult to understand the performance of neural networks. The well-known manifold hypothesis states that real-world high dimensional data (such as images) form lower-dimensional manifolds embedded in the high-dimensional space Carlsson et al. (2008), but these manifolds are tangled together and are difficult to separate. The classification process is then equivalent to stretching, squishing and separating the tangled manifolds apart. However, these operations pose a challenge: it is quite implausible that only affine transformations followed by pointwise nonlinear activations are sufficient to project or embed data into representative manifolds that are easily separable by class.

Therefore, instead of asking neural networks to separate the manifolds by a hyperplane or a surface, it is more reasonable to require points of the same manifold to be closer than points of other manifolds Olah (2014). Namely, the distance between manifolds of different classes should be large and the distance between manifolds of the same class should be small. This distance property is behind the concept of k-Nearest Neighbor (kNN) Cover & Hart (1967). Consequently, letting neural networks mimic kNN would combine the notion of manifolds with the desired distance property.

We explore kNN through two deep neural network models: sequence to sequence deep neural networks Sutskever et al. (2014) and memory networks Sukhbaatar et al. (2015). A family of our models are based on a sequence to sequence network. The new sequence to sequence model has the input sequence of length one corresponding to a sample, and then it decodes it to predict two sequences of output, which are the classes of closest samples and neighboring samples not necessarily in the training data, where we call the latter as out-of-sample feature vectors. We also propose a family of models built on a memory network, which has a memory that can be read and written to and is composed of a subset of training samples, with the goal of using it for predicting both classes of close samples and out-of-sample feature vectors. With the help of attention over memory vectors, our new memory network model generates the predicted label sequence and out-of-sample feature vectors. Both families of models use loss functions that mimic kNN. Computational experiments

show that the new sequence to sequence kNN model consistently outperforms benchmarks (kNN, a feed-forward neural network and a vanilla memory network). We postulate that this is due to the fact that we are forcing the model to 'work harder' than necessary (producing out-of-sample feature vectors).

Different from general classification models, our models predict not only labels, but also out-of-sample feature vectors. Usually a classification model only predicts labels, but as in the case of kNN, it is desirable to learn or predict the feature vectors of neighbors as well. Intuitively, if a deep neural network predicts both labels and feature vectors, it is forced to learn and capture representative information of input, and thus it should perform better in classification. Moreover, our models also function as synthetic oversamplers: we add the out-of-sample feature vectors and their labels (synthetic samples) to the training set. Experiments show that our sequence to sequence kNN model outperforms Synthetic Minority Over-sampling Technique (SMOTE) Chawla et al. (2002) and Adaptive Synthetic sampling (ADASYN) He et al. (2008) most of the times on imbalanced datasets.

Usually we allow models to perform kNN searching on the entire dataset, which we call the full versions of models, but kNN is computationally expensive on large datasets. We design an algorithm to resolve this and we test our models under such an 'out-of-core' setting: only a batch of data can be loaded into memory, i.e. kNN searching in the entire dataset is not allowed. For each such random batch, we compute the $K$ closest samples with respect to the given training sample. We repeat this $R$ times and find the closest $K$ samples among these $KR$ samples. These closest $K$ samples provide the approximate label sequence and feature vector sequence to the training sample based on the kNN algorithm. Computational experiments show that sequence to sequence kNN models and memory network kNN models significantly outperform the kNN benchmark under the out-of-core setting.

Our main contributions are as follows. First, we develop two types of deep neural network models which mimic the kNN structure. Second, our models are able to predict both labels of closest samples and out-of-sample feature vectors at the same time: they are both classification models and oversamplers. Third, we establish the out-of-core version of models in the situation where not all data can be read into computer memory or kNN cannot be run on the entire dataset. The full version of the sequence to sequence kNN models and the out-of-core version of both sequence to sequence kNN models and memory network kNN models outperform the benchmarks, which we postulate is because learning neighboring samples enables the model to capture representative features.

We introduce background and related works in Section 2, show our approaches in Section 3, and describe datasets and experiments in Section 4. Conclusions are in Section 5.

## 2 BACKGROUND AND LITERATURE REVIEW

There are several works trying to mimic kNN or applying kNN within different models. Mathy et al. (2015) introduced the boundary forest algorithm which can be used for nearest neighbor retrieval. Based on the boundary forest model, in Zoran et al. (2017), a boundary deep learning tree model with differentiable loss function was presented to learn an efficient representation for kNN. The main differences between this work and our work are in the base models used (boundary tree vs standard kNN), in the main objectives (representation learning vs classification and oversampling) and in the loss functions (KL divergence vs KL divergence components reflecting the kNN strategy and $L^2$ norm). Wang et al. (2017) introduced a text classification model which utilized nearest neighbors of input text as the external memory to predict the class of input text. Our memory network kNN models differ from this model in 1) the external memory: our memory network kNN models simply feed a random batch of samples into the external memory without the requirement of nearest neighbors and thus they save computational time, and 2) number of layers: our memory network kNN models have $K$ layers while the model proposed by Wang et al. (2017) has one layer. A higher-level difference is that Wang et al. (2017) considered a pure classification setting, while our models generate not only labels but out-of-sample feature vectors as well. Most importantly, the loss functions are different: Wang et al. (2017) used KL divergence as the loss function while we use a specially designed KL divergence and $L^2$ norm to force our models to mimic kNN.

The sequence to sequence model, one of our base models, has recently become the leading framework in natural language processing Sutskever et al. (2014) Cho et al. (2014). In Cho et al. (2014) an

RNN encoder-decoder architecture was used to deal with statistical machine translation problems. Sutskever et al. (2014) proposed a general end-to-end sequence to sequence framework, which is used as the basic structure in our sequence to sequence kNN model. The major difference between our work and these studies is that the loss function in our work forces the model to learn from neighboring samples, and our models are more than just classifiers - they also create out-of-sample feature vectors that improve accuracy or can be used as oversamplers.

In summary, the main differences between our work and previous studies are as follows. First, our models predict both labels of nearest samples and out-of-sample feature vectors rather than simply labels. Thus, they are more than classifiers: the predicted label sequences and feature vector sequences can be treated as synthetic oversamples to handle imbalanced class problems. Second, our work emphasizes on the out-of-core setting. All of the prior works related to kNN and deep learning assume that kNN can be run on the entire dataset and thus cannot be used on large datasets. Third, our loss functions are designed to mimic kNN, so that our models are forced to learn neighboring samples to capture the representative information.

SEQUENCE TO SEQUENCE MODEL

A family of our models are built on sequence to sequence models. A sequence to sequence (Seq2seq) model is an encoder-decoder model. The encoder encodes the input sequence to an internal representation called the 'context vector' which is used by the decoder to generate the output sequence. Usually, each cell in the Seq2seq model is a Long Short-Term Memory (LSTM) cell Hochreiter & Schmidhuber (1997) or a Gated Recurrent Unit (GRU) cell Cho et al. (2014).

Given input sequence $x_1, ..., x_T$, in order to predict output $Y_1^P, ..., Y_K^P$ (where the superscript $P$ denotes 'predicted'), the Seq2seq model estimates conditional probability $P(Y_1^P, ..., Y_t^P | x_1, ..., x_T)$ for $1 \leq t \leq K$. At each time step $t$, the encoder updates the hidden state $h_t^e$, which can also include the cell state, by $h_t^e = f_h^e(x_t, h_{t-1}^e)$ where $1 \leq t \leq T$. The decoder updates the hidden state $h_t^d$ by $h_t^d = f_h^d(Y_{t-1}^P, h_{t-1}^d, h_T^e)$ where $1 \leq t \leq K$. The decoder generates output $y_t$ by

$$y_t = g(Y_{t-1}^P, h_t^d, h_T^e), \tag{1}$$

and $Y_t^P = q(y_t)$ with $q$ usually being softmax function.

The model calculates the conditional probability of output $Y_1^P, ..., Y_K^P$ by

$$\Pr(Y_1^P, ..., Y_K^P | x_1, ..., x_T) = \prod_{t=1}^K \Pr(Y_t^P | Y_1^P, ..., Y_{t-1}^P).$$

END TO END MEMORY NETWORKS

The other family of our models are built on an end-to-end memory network (MemN2N). This model takes $x_1, ..., x_n$ as the external memory, a 'query' $x$, a ground truth $Y^{GT}$ and predicts an answer $Y^P$. It first embeds memory vectors $x_1, ..., x_n$ and query $x$ into continuous space. They are then processed through multiple layers to generate the output label $Y^P$.

MemN2N has $K$ layers. In the $t^{th}$ layer, where $1 \leq t \leq K$, the external memory is converted into embedded memory vectors $m_1^t, ..., m_n^t$ by an embedding matrix $A^t$. The query $x$ is also embedded as $u^t$ by an embedding matrix $B^t$. The attention scores between embedded query $u^t$ and memory vectors $(m_i^t)_{i=1,2,...,n}$ are calculated by $p^t = softmax((u^t)^T m_1^t, (u^t)^T m_2^t, ..., (u^t)^T m_n^t)$. Each $x_i$ is also embedded to an output representation $c_i^t$ by another embedding matrix $C^t$. The output vector from the external memory is defined as $o^t = \sum_{i=1}^n p_i^t c_i^t$. By a linear mapping $H$, the input to the next layer is calculated by $u^{t+1} = Hu^t + o^t$. Sukhbaatar et al. (2015) suggested that the input and output embeddings are the same across different layers, i.e. $A^1 = A^2 = ... = A^K$ and $C^1 = C^2 = ... = C^K$.

In the last layer, by another embedding matrix $W$, MemN2N generates a label for the query $x$ by $Y^P = softmax(W(Hu^K + o^K))$.

## 3   kNN MODELS

Our sequence to sequence kNN models are built on a Seq2seq model, and our memory network kNN models are built on a MemN2N model. Let $K$ denote the number of neighbors of interest.

*Vector to Label Sequence (V2LS) Model*

Given an input feature vector $x$, a ground truth label $Y^{GT}$ (a single class corresponding to $x$) and a sequence of labels $Y_1^T, Y_2^T, ..., Y_K^T$ corresponding to the labels of the $1^{st}, 2^{nd}, ..., K^{th}$ nearest sample to $x$ in the entire training set, V2LS predicts a label $Y^P$ and $Y_1^P, Y_2^P, ..., Y_K^P$, the predicted labels of the $1^{st}, 2^{nd}, ..., K^{th}$ nearest samples. Since $Y_1^T, Y_2^T, ..., Y_K^T$ are obtained by using kNN upfront, the real input is only $x$ and $Y^{GT}$.

In the V2LS model, by a softmax operation with temperature after a linear mapping $(W_y, b_y)$, the label of the $t^{th}$ nearest sample to $x$ is predicted by $Y_t^P = softmax((W_y y_t + b_y)/\tau)$ where $y^t$ is as in (1) for $t = 1, 2, ..., K$ and $\tau$ is the temperature of softmax. By taking the average of predicted label distributions, the label of $x$ is predicted by $Y^P = \sum_{t=1}^{K} Y_t^P / K$. Temperature $\tau$ controls the "peakedness" of $Y_t^P$. Values of $\tau$ below 1 push $Y_t^P$ towards a Dirac distribution, which is desired in order to mimic kNN. We design the loss function as

$$L_1 = \mathbb{E}\{\sum_{t=1}^{K} D_{KL}(Y_t^T || Y_t^P)/K + \alpha D_{KL}(Y^{GT} || Y^P)\}$$

where the first term captures the label at the neighbor level, the second term for the actual ground truth and $\alpha$ is a hyperparameter to balance the two terms. The expectation is taken over all training samples, and $D_{KL}$ denotes the Kullback-Leibler divergence. Due to the fact that the first term is the sum of KL divergence between predicted labels of nearest neighbors and target labels of nearest neighbors, it forces the model to learn information about the neighborhood. The second term considers the actual ground truth label: a classification model should minimize the KL divergence between the predicted label (average of $K$ distributions) and the ground truth label. By combining the two terms, the model is forced to not only learn the classes of the final label but also the labels of nearest neighbors.

In inference, given an input $x$, V2LS predicts $Y^P$ and $Y_1^P, Y_2^P, ..., Y_K^P$, but only $Y^P$ is the actual output; it is used to measure the classification performance.

*Vector to Vector Sequence (V2VS) Model*

We use the same structure as the V2LS model except that in this model, the inputs are a feature vector $x$ and a sequence of feature vectors $X_1^T, X_2^T, ..., X_K^T$ corresponding to the $1^{st}, 2^{nd}, ..., K^{th}$ nearest sample to $x$ among the entire training set (calculated upfront using kNN). V2VS predicts $X_1^P, X_2^P, ..., X_K^P$ which denote the predicted out-of-sample feature vectors of the $1^{st}, 2^{nd}, ..., K^{th}$ nearest sample. Since $X_1^T, X_2^T, ..., X_K^T$ are obtained using kNN, this is an unsupervised model.

The output of the $t^{th}$ decoder cell $y_t$ is processed by a linear layer $(W_{x1}, b_{x1})$, a $ReLU$ operation and another linear layer $(W_{x2}, b_{x2})$ to predict the out-of-sample feature vector $X_t^P = W_{x2} max\{W_{x1} y_t + b_{x1}, 0\} + b_{x2}, t = 1, 2, ..., K$. Numerical experiments show that $ReLU$ works best compared with $tanh$ and other activation functions. The loss function is defined to be the sum of $L^2$ norms as

$$L_2 = \mathbb{E}\{\sum_{t=1}^{K} ||X_t^P - X_t^T||^2\}.$$

Since the predicted out-of-sample feature vectors should be close to the input vector, learning nearest vectors forces the model to learn a sequence of approximations to something very close to the identity function. However, this is not trivial. First it does not learn an exact identity function, since the output is a sequence of nearest neighbors to input, i.e. it does not simply copy the input $K$ times. Second, by limiting the number of hidden units of the neural network, the model is forced to capture the most representative and condensed information of input. A large amount of studies

have shown this to be beneficial to classification problems Erhan et al. (2010)Vincent et al. (2010)He et al. (2016).

In inference, we predict the label of $x$ by finding the labels of out-of-sample feature vectors $X_t^P$ and then perform majority voting among these $K$ labels.

*Vector to Vector Sequence and Label Sequence (V2VSLS) Model*

In previous models, V2LS learns to predict labels of nearest neighbors and V2VS learns to predict feature vectors of nearest neighbors. Combining V2LS and V2VS together, this model predicts both $X_t^P$ and $Y_t^P$. The loss function is a weighted sum of the two loss functions in V2LS and V2VS: $L = L_1 + \lambda L_2$, where $\lambda$ is a hyperparameter to account for the scale.

The $L^2$ norm part enables the model to learn neighboring vectors. As discussed in the V2VS model, this is beneficial to classification since it drives the model to capture representative information of input and nearest neighbors. The $KL$ part of the loss function focuses on predicting labels of nearest neighbors. As discussed in the V2LS model, the two terms in the $KL$ loss force the model to learn both neighboring labels and the ground truth label. Combining the two parts, the V2VSLS model is able to predict nearest labels and out-of-sample feature vectors, as well as one final label for classification. In inference, given an input $x$, V2VSLS generates $Y^P$, $X_1^P, X_2^P, ..., X_K^P$ and $Y_1^P, Y_2^P, ..., Y_K^P$. Still only $Y^P$ is used in measuring classification performance of the model.

*Memory Network - kNN (MNkNN) Model*

The MNkNN model is built on the MemN2N model, which has $K$ layers stacked together. After these layers, the MemN2N model generates a prediction. In order to mimic kNN, our MNkNN model has $K$ layers as well but it generates one label after each layer, i.e. after the $t^{th}$ layer, it predicts the label of the $t^{th}$ nearest sample. It mimics kNN because the first layer predicts the label of the first closest vector to $x$, the second layer predicts the label of the second closest vector to $x$, etc.

This model takes a feature vector $x$, its ground truth label $Y^{GT}$, a random subset $x_1, x_2, ..., x_n$ from the training set (to be stored in the external memory) and $Y_1^T, Y_2^T, ..., Y_K^T$ denoting the labels of the $1^{st}, 2^{nd}, ..., K^{th}$ nearest samples to $x$ among the entire training set (calculated upfront using kNN). It predicts a label $Y^P$ and a sequence of $K$ labels of closest samples $Y_1^P, Y_2^P, ..., Y_K^P$.

After the $t^{th}$ layer, by a softmax operation with temperature after a linear mapping $(W_y, b_y)$, the model predicts the label of $t^{th}$ nearest sample by $Y_t^P = softmax((W_y(Hu^t + o^t) + b_y)/\tau)$ where $t = 1, 2, ..., K$. The role of $\tau$ is the same as in the V2LS model. Taking the average of the predicted label distributions, the final label of $x$ is calculated by $Y^P = \sum_{t=1}^{K} Y_t^P / K$. Same as V2LS, the loss function of MNkNN is defined as

$$\overline{L_1} = \mathbb{E}\{\sum_{t=1}^{K} KL(Y_t^T || Y_t^P)/K + \alpha KL(Y^{GT} || Y^P)\}.$$

The first term accounts for learning neighboring information, and the second term forces the model to provide the best single candidate class.

In inference, the model takes a query $x$ and random samples $x_1, x_2, ..., x_n$ from the training set, and generates the predicted label $Y^P$ as well as a sequence of nearest labels $Y_1^P, Y_2^P, ..., Y_K^P$.

*Memory Network - kNN with Vector Sequence (MNkNN_VEC) Model*

This model is built on MNkNN, but it predicts out-of-sample feature vectors $X_t^P$ as well. By a linear mapping $T$, a $ReLU$ operation and another linear mapping $(W_x, b_x)$, the feature vectors are calculated by $X_t^P = W_x max\{T(Hu^t + o^t), 0\} + b_x$. Same as the V2VSLS model, combining the $L^2$ norm and the KL divergence together, the loss function is defined as

$$\overline{L} = \overline{L_1} + \lambda \mathbb{E}\{\sum_{t=1}^{K} ||X_t^P - X_t^T||^2\}$$

As discussed in the V2VSLS model, having such loss function forces the model to learn both the feature vectors and the labels of nearest neighbors.

*Out-of-Core Models*

In the models exhibited so far, we assume that kNN can be run on the entire dataset exactly to compute the $K$ nearest feature vectors and corresponding labels to an input sample. However, there are two problems with this assumption. First, this can be very computationally expensive if the dataset size is large. Second, the training dataset might be too big to fit in memory. When either of these two challenges is present, an out-of-core model assuming it is infeasible to run a 'full' kNN on the entire dataset has to be invoked. The out-of-core models avoid running kNN on the entire dataset, and thus save computational time and resources.

Let $B$ be the maximum number of samples that can be stored in memory, where $B > K$. For a training sample $x$, we sample a subset $S$ from the training set (including $x$) where $|S| = B$, then we run kNN on $S$ to obtain the $K$ nearest feature vectors and corresponding labels to $x$, which are denoted as $Y^T(S) = \{Y_1^T, Y_2^T, ..., Y_K^T\}$ and $X^T(S) = \{X_1^T, X_2^T, ..., X_K^T\}$ for $x$ in the training process. The previously introduced loss functions $L$ and $\overline{L}$ depend on $x, Y^{GT}, X^T(S), Y^T(S)$ and the model parameters $\Theta$, and thus our out-of-core models are to solve

$$\min_{\Theta} \mathbb{E}_x \, \mathbb{E}_S \{\widetilde{L}(x, Y^{GT}, X^T(S), Y^T(S), \Theta)\}$$

where $\widetilde{L}$ is either $L$ or $\overline{L}$.

Sampling a set of size $B$ and then finding the nearest $K$ samples only once, however, are insufficient on imbalanced datasets, due to the low selection probability for minor classes. To resolve this, we iteratively take $R$ random batches: each time a random batch is taken, we update the closest samples $X^T(S)$ by the $K$ closest samples among the current batch and the $K$ previous closest samples. These resulting nearest feature vectors and corresponding labels are used as $X^T(S)$ and $Y^T(S)$ for $x$ in the loss function. Note that we allow the previously selected samples to be selected in later sampling iterations. The entire algorithm is exhibited in Algorithm 1.

---

**ALGORITHM 1:** Out-of-core framework

**for** *epoch = 1,...,T* **do**
    **for** *training sample $x$* **do**
        Let $X^T = \varnothing$;
        **for** *r = 1 to R* **do**
            Randomly draw $B$ samples from training set;
            $U$ = nearest $K$ samples to $x$ among the $B$ samples;
            Let $X^T$ be the nearest $K$ samples to $x$ among $U \cup X^T$;
        **end**
        Update parameters by a gradient iteration: $\Theta^R = \Theta^R - \alpha\nabla\widetilde{L}(x, Y^{GT}, X^T, Y^T, \Theta^R)$;
    **end**
**end**

---

# 4   COMPUTATIONAL EXPERIMENTS

Four classification datasets are used: Network Intrusion (NI) Hettich & Bay (1999), Forest Cover-type (COV) Blackard & Dean (1998), SensIT Duarte & Hu (2004) and Credit Card Default (CCD) Yeh & hui Lien (2009). Details of these datasets are in Table 1. We only consider 3 classes in NI and COV datasets due to significant class imbalance. All of the models have been developed in Python 2.7 by using Tensorflow 1.4.

For each dataset we experiment with 5 different seeds. F-1 score is used as the performance measure and all reported numbers are averages taken over 5 random seeds. We discuss the performance of the models in two aspects: classification and oversampling.

Table 1: Datasets information.

|  | NI | COV | SensIT | CCD |
|---|---|---|---|---|
| Dataset Size | 796,497 | 530,895 | 98,528 | 30,000 |
| Feature Size | 41 | 54 | 100 | 23 |
| Number of Classes | 3 | 3 | 3 | 2 |

CLASSIFICATION

As comparisons against memory network kNN models and sequence to sequence kNN models, we use kNN with Euclidean metric, a 4-layer feed-forward neural network (FFN) trained using the Adam optimization algorithm (which has been calibrated) and MemN2N (since MNkNN and MNkNN_VEC are built on MemN2N) as three benchmarks. Value $K = 5$ is used in all models because it yields the best performance with low standard deviation among $K = 3, 4, ..., 10$. Increasing $K$ beyond $K = 5$ is somewhat detrimental to the F-1 scores while significantly increasing the training time.

In the sequence to sequence kNN models, LSTM cells are used. In the memory network kNN models, the size of the external memory is 64 since we observe that models with memory vectors of size 64 generally provide the best F-1 scores with acceptable running time. Both sequence to sequence kNN models and memory network kNN models are trained using the Adam optimization algorithm with initial learning rate set to be 0.01. Dropout with probability 0.2 and batch normalization are used to avoid overfitting. Regarding the choices of other hyperparameters, we find that $\tau = 0.85$, $\lambda = 0.12$ and $\alpha = 9.5$ provide overall the best F-1 scores.

We first discuss the full models that can handle all of the training data, i.e. kNN can be run on the entire dataset. Figure 1 show that in the full model case, V2VSLS consistently outperforms other models on all four datasets. t-tests show that it significantly outperforms three benchmarks at the 5% level on all four datasets. Moreover, it can also be seen that predicting not only labels but feature vectors as well is reasonable, since V2VSLS consistently outperforms V2LS and MNkNN_VEC consistently outperforms MNkNN. Models predicting feature vectors outperform models not predicting feature vectors on all datasets. These memory based models exhibit subpar performance, which is expected since they only consider 64 training samples at once (despite using exact labels). From Figure 1 we also observe that standard deviations do not differ significantly.

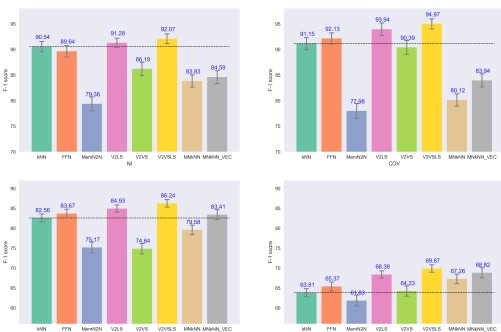

Figure 1: Full model F-1 score. Numbers above bars denote the average F-1 scores. The error bars denote the standard deviations. Note that the y-axis does not start from 0.

In the out-of-core versions of our models, $R$ is set to be 50, since we observe that increasing 50 to, for instance, 100, only has a slight impact on F-1 scores. However, increasing $R$ from 50 to 100 substantially increases the running time. The batch size $B$ of the out-of-core models is set to be 64 since it is found to provide overall the best F-1 scores with reasonable running time.

Figure 2 shows the results of our models under the out-of-core assumption when $R = 50$ and $B = 64$. The comparison shows that both V2VLSL and MNkNN_VEC significantly outperform the kNN benchmark based on t-tests at the 5% significance level. The kNN benchmark provides a low score since we restrict the batch size (or memory size) to be 64, and it turns out that kNN is substantially affected by the randomness of batches. Our models (except V2VS, since it makes

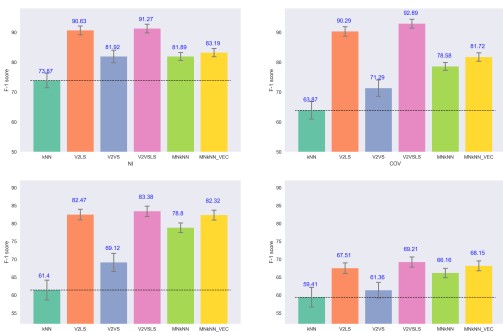

Figure 2: Out-of-core model with R=50. Note that the y-axis does not start from 0.

Table 2: Full model and out-of-core (OOC) model comparison on SensIT.

|  | kNN | V2LS | V2VS | V2VSLS | MNkNN | MNkNN_VEC |
|---|---|---|---|---|---|---|
| Full F-1 | 82.56 | 84.93 | 74.84 | 86.24 | 79.58 | 83.41 |
| OOC F-1 | 61.40 | 82.47 | 69.12 | 83.38 | 78.80 | 82.32 |
| Full time (s) | 312 | 443+635 | 857+1358 | 1391+1802 | 443+692 | 1391+1081 |
| OOC time (s) | 193 | 287+619 | 488+1316 | 741+1846 | 287+703 | 741+1055 |

predictions only depending on feature vector sequences) are robust under the out-of-core setting, because the weight of the ground truth label in the loss function is relatively high so that even if the input nearest sequences are noisy, they still can focus on learning the ground truth label and making reasonable predictions.

Table 2 shows a comparison between the full and out-of-core models with $R = 50, B = 64$ on the SensIT dataset. The running time of our models are broken down to two parts: the first part is the time to obtain sequences of $K$ nearest feature vectors and labels and the second part is the model training time. Under the out-of-core setting, overall the kNN sequence preprocessing time is saved by approximately 40% while the models perform only slightly worse.

OVERSAMPLING

Since V2VSLS and MNkNN_VEC are able to predict out-of-sample feature vectors, we also regard our models as oversamplers and we compare them with two widely used oversampling techniques: SMOTE and ADASYN. We only test V2VSLS since it is the best model that can handle all of the data. In our experiments, we first fully train the model. Then for each sample from the training set, V2VSLS predicts $K = 5$ out-of-sample feature vectors which are regarded as synthetic samples. We add them to the training set if they are in a minority class until the classes are balanced or there are no minority training data left for creating synthetic samples. In our oversampling experiments, we use $\lambda = 1.3$ and $\alpha = 3$.

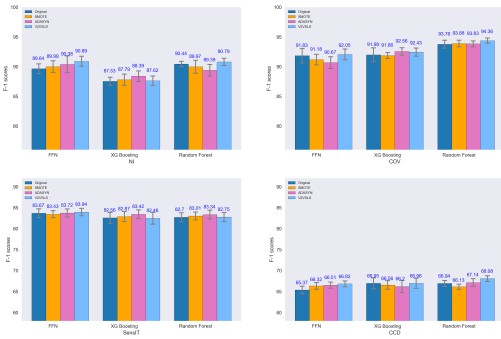

Figure 3: Oversampling: F-1 score comparison.

Table 3: Oversampling techniques comparison.

|  | NI | COV | SensIT | CCD |
|---|---|---|---|---|
| Best model | FFN+V2VSLS | RF+V2VSLS | FFN+V2VSLS | RF+V2VSLS |
| Best F-1 score | 90.89 | 94.36 | 83.92 | 68.08 |
| Better than best SMOTE by | 1% | 0.51% | 0.61% | 2.28% |
| Better than best ADASYN by | 0.6% | 0.56% | 0.26% | 1.4% |

Figure 3 shows the F-1 scores of FFN, extreme gradient boosting (XGB) and random forest (RF) classification models, with different oversampling techniques, namely, original training set without oversampling, SMOTE, ADASYN and V2VSLS. V2VSLS performs the best among all combinations of classification models and oversampling techniques, as shown in Table 3. Although most of the time models on datasets with three oversampling techniques outperform models on datasets without oversampling, the classification performance still largely depends on the classification model used and which dataset is considered.

Figure 4 shows a t-SNE van der Maaten & Hinton (2008) visualization of the original set and the oversampled set, using SensIT dataset, projected onto 2-D space. Although SMOTE and ADASYN overall perform well, their class boundaries are not as clean as those obtained by V2VSLS.

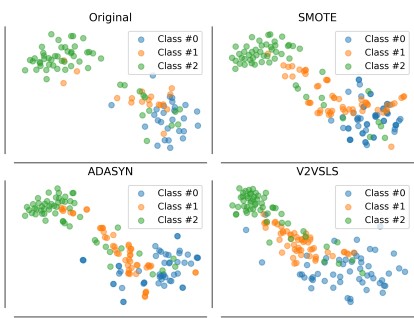

Figure 4: t-SNE visualization of different oversampling methods

SUMMARY

In summary, we find that it is beneficial to have neural network models learn not only labels but feature vectors as well. In the full models, V2VSLS outperforms all other models consistently; in the out-of-core models, both V2VSLS and MNkNN_VEC significantly outperform the kNN benchmark. As an oversampler, the average F-1 score based on the training set augmented by V2VSLS outperforms that of SMOTE and ADASYN.

We recommend to run V2VSLS with large $\alpha$ and small $\lambda$ for classification. In the oversampling scenario, however, we suggest to use small $\alpha$ and large $\lambda$ so that the model focuses more on the feature vectors, i.e. synthetic samples.

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
