# OpenReview forum: "k-Nearest Neighbors by Means of Sequence to Sequence Deep Neural Networks and Memory Networks"
_ICLR.cc/2019/Conference_

### Official Review · AnonReviewer3 · 2018-11-01
**Combine KNN of each training data point into the neural network models, but the motivation does not make sense.**

**Rating:** 4
**Confidence:** 4

**Review:**

To exploit the near neighbor/manifold features, this paper proposes to combine k-nearest neighbors of each training data point into the neural network models.  Specifically, the authors propose two families of models built on the popular sequence to sequence neural network models and memory network models, which mimic the k-nearest neighbors model in model learning. Besides, the final label of the classification task will be learned, a sequence of nearest neighbor labels and a sequence of out-of-sample feature vectors (for oversampling) will be also learned in the same time, similar with the multi-task approaches. Since the proposed models are based k-nearest neighbor calculations, which is time-consuming, they also design an algorithm for the ‘out-of-core’ situation, say load a small portion of data each time to approximately calculate the neighbors. Experiments show that some proposed models work better than baselines in classification and oversampling.
Strong points:
(1) As similar with the multi-task setting, the proposed model can output some side useful results, such as oversampling vectors.
(2) The proposed models work well on the ‘out-of-core’ situation, which shows that the models are robust.
Concerns or suggestions:
(1) The training data $x$ is just one data point, it is not a sequence of data. So the idea to model it in a sequence to sequence setting does not make sense.
(2) K-nearest neighbors are a set but not a sequence. To model them as a sequence is also strange. The i-th nearest neighbor does not necessarily dependent on the i-1-th nearest neighbor. For example, we consider the one-dimensional case, the focus data may lie between its first and second nearest neighbors. In this case, there is no clear sequence dependence from the second neighbor to the first neighbor.
(3) The experiments are not sufficient. They only compare with some weak baselines, such as KNN. As the classification task, there are many state-of-the-art models. Besides of these standard classification models, we strongly suggest comparing with the previous method, Wang et al. (2017), which also proposes to combine the k-nearest neighbors into memory network models. I am surprised that the authors did not compare with this very related work. In my opinion, the idea of utilizing nearest neighbors as external memory in Wang et al. (2017) makes more senses.
(4) The experimental results of some proposed sub-models (key parts of final models) are even worse than the basic kNN model. I should say that the results are not good enough to support the proposed methods.

---

> ### Author Response · Authors · 2018-11-14
> **Response3**
>
> Thank you very much for the comments. Regarding the first point, we can regard the input as the sequence of length 1, and then the seq2seq model can be applied in this situation.
>
> Regarding the second point, an order is quite natural – based on the distance from the sample. In summary, while there is no direct order, the order based on the distance makes sense in our opinion. When predicting the label of a sample, the model tries to predict $k$ probability distributions and mimic “majority voting.” We have also tried changing the loss function to be less (or not) dependent on the order of the feature vector sequence: we tried changing the sum of MSE losses (Sum (X^T_i – X^P_i)^2) to the MSE loss between every X^P_i and the average feature vector (Sum (X^T_i)/k) or the feature vector X itself. The experiments show that our current loss function (dependent sequence) outperforms these options (independent or less dependent sequence). To get a better handle on the order, we are currently conducting an experiment where we swap some elements in the order. We will post here the findings as soon as we have them.
>
> [UPDATE] To get a better handle on the order of nearest neighbors, we have tried to arbitrarily swap the 1st and the 3rd element in the order, and the results only have a tiny difference. The results are shown below (4 datasets):
> V2VSLS: 92.07/94.97/86.24/69.87
> Swapped: 91.79/94.56/85.99/69.42
> These numbers are based on the average of five runs. The results suggest that the order of nearest neighbors does not have a big impact. Note also that in our loss function there is a KL divergence term between the ground truth label and the final predicted label with a high weight.
>
> Regarding the third point, in our results, our model outperforms a fine-tuned neural network (with regularization, using the Adam optimization algorithm, etc.), random forest, XGBoosting etc., which we consider as the state-of-the-art general classifiers. For example, XGBoosting is a frequent winner in many Kaggle competitions.
>
> [UPDATE] We have also implemented the algorithm from Wang et al (2017), which also utilized the nearest neighbors to make predictions. The comparison is shown below (4 datasets):
> V2VSLS: 92.07/94.97/86.24/69.87
> Memory Network: 79.36/77.98/75.17/61.83
> Wang: 64.18/69.64/54.29/52.18
> In the implementation of Wang, we have fine-tuned the hyperparameters: K, I, Learning rate. The optimizer used is Adam. These numbers are based on the average of five runs. There is a gap between our models and Wang’s model, and we were unable to further improve the Wang's model.
>
> Regarding the fourth point, I assume you were referring to the memory networks models and V2VS. That is correct – we wanted to assess whether those different structures would improve the performance, and unfortunately, they do not perform well. We are reporting these results in order for other researchers not to spend time on such attempts. However, the V2VSLS performance is excellent, and therefore V2VSLS is the main method our work suggests.

---

### Official Review · AnonReviewer2 · 2018-11-02
**need more clear writing and strong experimental results**

**Rating:** 5
**Confidence:** 4

**Review:**

I had a hard time understanding this paper. The approach is clearly about combining kNN with neural networks, but it wasn’t clear how it is done. After reading the whole paper, my guess is that kNN is done on raw data first, and then its results are used for training a neural network. In particular, a network is trained to predict the labels of neighboring samples, which are obtained by kNN beforehand. A simple figure explaining it in the introduction would be very helpful since the idea is not that complex.

Also, the authors also fail to give an adequate explanation on why the method works. The only reason I can think of is that this regularization forces the model to detect if a sample near a class boundary. This is because when a sample is far from boundaries and surrounded by samples of the same class, the model would simply predict that class label. The same is true when predicting out-of-sample vectors because the average position of K'th neighbor is likely to overlap with the input sample due to the randomness of sampling.

I don’t really see why a memory-based model is introduced. The external memory is used for holding random samples. It is not clear how the model can use such random samples for making predictions. Also, the authors give no explanation to why it should help. The results also don’t show the benefit of a memory-based model. Maybe the authors should look into models that output a set instead of a sequence since neighbors are more like a set in their structure.

The experimental results show clear improvements over basic baselines, so the method is doing some regularization. However, I'm not very familiar with datasets used here and their state-of-art. They are relatively low dimensional compared to usual datasets used in deep learning. It is not clear if the method can scale to high dimensional data such as images. The vanilla neural network is not really a strong baseline here. Since the authors proposed a regularization technique, it should be compared with other regularization techniques in neural networks.

Pros:
- a simple idea
- encouraging experimental results

Cons:
- confusing read
- no clear intuition is given
- restricted to low-dimensional datasets
- strong baselines needed
- the plots are too small to see (impossible to see when printed)

Other comments:
- The authors are using the term "feature vector" to refer to a data point. However, in the context of neural networks, "feature vector" often means a hidden representation of a neural network.
- why repeat "randomly draw B samples" R times? why not directly sample RxB samples?
- "it is quite implausible that only affine ..." any evidence to support this?
- The model is not really "sequence-to-sequence" since the input is not a sequence.

---

> ### Author Response · Authors · 2018-11-14
> **Response2**
>
> Thank you very much for the comments. Your understanding is correct – kNN is done on raw data first, then the results are used for training a neural network. Regarding the reason of why the method works, we postulate that this is because the model is forced to learn more than just the final label of a sample; it also needs to learn its nearest feature vectors. The idea is similar to an autoencoder, which is able to compress and then recover the data. By projecting the data to a lower dimensional space, the model is forced to learn the most representative information of inputs, so that the compressed information can be used to recover the original data. If the model is forced to reconstruct more complex information from compressed information, it is expected that it needs to work harder and thus to produce more exact predictions.
>
> Regarding the reason why we consider a memory network, the structure of a vanilla memory network fits perfectly with our original thought: after each hop of a memory network, the network outputs a label or a feature vector. This is very similar to the sequence to sequence concept. However, experiments show that the memory network family does not perform well – the sequence to sequence family works much better. We wanted to assess whether those different structures would improve the performance, and unfortunately, they do not perform well. We are reporting these results in order for other researchers not to spend time on such attempts.
>
> Regarding  the comment about baseline models, we have fine-tuned the vanilla neural network with dropout regularization. The baseline result in the paper already includes this regularization. We have also tried the L1/L2 regularization, but dropout outperforms them on the baseline neural network model.
>
> For the point ‘the model is not really a sequence to sequence model since the input is not a sequence,’ we can in fact regard the input as the sequence of length 1, and then the seq2seq model can be applied in this situation.
>
> For the point ‘"it is quite implausible that only affine ..." any evidence to support this,’ for example the two linking rings are not easy to separate if we only use affine transformation etc.
>
> Regarding ‘why repeat "randomly draw B samples" R times? why not directly sample RxB samples,’ our assumption is that at most B samples can fit to computer memory (and thus RxB cannot). Randomly drawing B samples R times fixes this problem.
>
> [UPDATE] To get a better handle on the order of nearest neighbors, we have tried to arbitrarily swap the 1st and the 3rd element in the order, and the results only have a tiny difference. The results are shown below (4 datasets):
> V2VSLS: 92.07/94.97/86.24/69.87
> Swapped: 91.79/94.56/85.99/69.42
> These numbers are based on the average of five runs. The results suggest that the order of nearest neighbors does not have a big impact. Note also that in our loss function there is a KL divergence term between the ground truth label and the final predicted label with a high weight.
>
> [UPDATE] We have also implemented the algorithm from Wang et al (2017), which also utilized the nearest neighbors to make predictions. The comparison is shown below (4 datasets):
> V2VSLS: 92.07/94.97/86.24/69.87
> Memory Network: 79.36/77.98/75.17/61.83
> Wang: 64.18/69.64/54.29/52.18
> In the implementation of Wang, we have fine-tuned the hyperparameters: K, I, Learning rate. The optimizer used is Adam. These numbers are based on the average of five runs. There is a gap between our models and Wang’s model, and we were unable to further improve the Wang's model.

---

> > ### Comment · AnonReviewer2 · 2018-11-30
> > **still need more works**
> >
> > I increased my score since I do agree that "it also needs to learn its nearest feature vectors" is an interesting approach and the experimental results support it. However, I still think the paper needs more work. The memory network wasn't an appropriate model to use here. Even seq2seq model is not a perfect fit since neighbors don't have sequential structure. I think the new swap experiment confirms that. I would suggest to use set models (deepset, transformer) instead. Also, I think more investigation is needed to understand why the method works.

---

> > > ### Author Response · Authors · 2018-12-06
> > > **Response 2**
> > >
> > > Thank you for the response. Our intuition is that in classification tasks, the distance between manifolds of different classes should be large and the distance between manifolds of the same class should be small. Therefore, letting neural networks mimic kNN would combine neural networks with the desired distance property of kNN. Our model learns the class label distribution and puts regularizations on it by means of nearest neighbor label distributions & features.
> > >
> > > In such a situation, the order of the nearest neighbors should matter, but should not be very crucial since they are regularizations. We have completed another experiment where we swap the order of the first nearest neighbor and the last nearest neighbor (F-1 score in each dataset is split by “/” below):
> > >
> > > Original V2VSLS: 92.07/94.97/86.24/69.87
> > > Swapped: 91.15/93.81/85.35/68.44
> > >
> > > There is a consistent decrease of F-1 between the original model and the model where we swap the first NN and the last NN. We claim that the order of NN matters, but not drastically. This is due to the loss function having a KL divergence term between the ground truth label and the final predicted label with a high weight, but the weights of regularizations are much smaller so that they have a minor effect.

---

### Official Review · AnonReviewer1 · 2018-11-02
**Training to predict kNN features and labels improves generalization.**

**Rating:** 6
**Confidence:** 4

**Review:**

This work uses sequence-to-sequence and memory network neural nets to learn a
network that not only predicts a label, but also predicts nearest neighbors and
their label.  The intuition is that by training on a related but harder task,
the network is forced to learn not just about sampled points but about the
behavior in regions around the actual training examples.

Their work shows that imposing these additional requirements on the model does
result in better performance on unseen data, where only the label of the unseen
data point is required as output.  They show that learning more about the
global {feature,label} distribution improves F1 scores, and suggest that their
methods can be used as an example generator for datasets with class imbalance.

The writing was clear.  I had only 1 misunderstanding that cause me trouble,
namely the first sentence of "Classification", where I might put the word
'benchmark' up front rather than at the end of this somewhat long sentence.
By the time 'knn benchmark' appeared some paragraphs later, I realized I
had missed something, and had to backtrack.

Figs 1--3 are only viewable on screen with fairly extreme magnification. I
found different viewers varied in legibility at these extreme zoom settings.
However, when expanded so y-axis numbers could be deciphered, it turned out
that actual improvements in F1 score were rather modest.

For 4 datasets, their F1 scores of their best method, V2VSLS, improved by ~ 1.5
to 6%.  They found that their methods were less affected by out-of-core
computation than the benchmark kNN F1 scores.

I did have some questions about the problem formulation.  I did not really
understand why the models wanted to reproduce the exact ordering of the
nearest neighbors, other than this is easy to formulate.  This makes sense
for n.n. label sequences, where further points might be in some neighboring
class.  But for predicting the feature vectors of n.n., it seems that the
exact order is not robust, in the sense very small distance changes can cause
abrupt shifts in the target nearest-neghbor ordering.  Is there some way
for me to understand why this does not pose a problem?  Or is there perhaps
a way to make the loss function a bit less dependent on the precise order
of the generated feature vector sequence?

In the computational experiments, with the choice of datasets, it was often
hard for me to judge how much different aspects of their 4 network structures
were really being excercised.  The main issue is the all problems used only
2--3 classes.  I could not guess what fraction of data points had actual label
changes within the 5 n.n..  For many datasets, "same class" might be a pretty
good predictor of nearest neighbor label.

Alternatively, one can consider the other extreme, of very many classes.
Does it still make sense to try to predict the order of nearest neighbor labels in
such a setting?

The authors provide evidence of some performance improvement, but I would
encourage them to provide some intuition about what the networks are actually
learning.  Some of this can be done with their existing data.

For example, on average, what are the distances from predicted feature vectors
to the actual nn feature vectors?  If the feature sequence is typically badly
predicted, then this might allow the authors to propose that the network is
*actually* learning some simpler features of the the distributions underlying
an actual {feature,label} sequence.  This might allow simplified training losses,
based, on things like direction and distance to same-label cluster center,
direction+distance to average same-class nn.s, direction and distance to
closest differently labeled cluster, etc.  Or does their data suggest that
their models are actually learning the precise nearest neighbor ordering?

Such considerations might be able to improve the OOC training, since
"global" aspects of the distribution features (like "cluster center")
remain approx. valid as training batches changes.

The other question I had was with the oversampling proposition.  The extent
of class imbalance in the datasets is not described.  Perhaps it belongs in
Table 1.  Their approach seems a lot of work for modest gains usually available
with oversampling.  Can the authors provide any guideline for how many members a
minority class should have before using their sequence-to-sequence technique?

Pros: they improve generalization to unseen data.
Cons: their models are considerably more complex, and they do not analyze their
data in enough detail to suggest whether their complexity is necessary, or perhaps
could be reduced.  Figures are too small (many unreadable in printed copy).
Datasets have very few classes and the extent to which nearest neighbors are
of different class not reported.

---

> ### Author Response · Authors · 2018-11-14
> **Response1**
>
> Thank you very much for the comments. For the issue of ordering of nearest neighbors, we postulate that this is because when predicting the label of a sample, the model tries to predict k probability distributions and mimic “majority voting.” For example, for a sample x, the predicted label sequence is, say, [2,1,1,1,2], the final predicted label would very likely be the same if the predicted label sequence is [1,2,1,1,2], due to “majority voting.” An order is quite natural – based on the distance from the sample. In summary, while there is no direct order, the order based on the distance makes sense in our opinion. To get a better handle on the order, we are currently conducting an experiment where we swap some elements in the order. We will post here the findings as soon as we have them.
>
> [UPDATE] We have tried to swap the 1st and the 3rd element in the order, and the results only have a tiny difference. The results are shown below (4 datasets):
> V2VSLS: 92.07/94.97/86.24/69.87
> Swapped: 91.79/94.56/85.99/69.42
> These numbers are based on the average of five runs. The results suggest that the order of nearest neighbors does not have a big impact. Note also that in our loss function there is a KL divergence term between the ground truth label and the final predicted label with a high weight.
>
> [UPDATE] We have also implemented the algorithm from Wang et al (2017), which also utilized the nearest neighbors to make predictions. The comparison is shown below (4 datasets):
> V2VSLS: 92.07/94.97/86.24/69.87
> Memory Network: 79.36/77.98/75.17/61.83
> Wang: 64.18/69.64/54.29/52.18
> In the implementation of Wang, we have fine-tuned the hyperparameters: K, I, Learning rate. The optimizer used is Adam. These numbers are based on the average of five runs. There is a gap between our models and Wang’s model, and we were unable to further improve the Wang's model.
>
> Regarding your suggestion of changing the loss function to be less dependent on the order of the feature vector sequence, we have actually tried to change the sum of MSE losses (Sum (X^T_i – X^P_i)^2) to the MSE loss between every X^P_i and the average feature vector (Sum (X^T_i)/k) or the feature vector X itself. The experiments have showed that our current loss function outperforms these options.
>
> Regarding the second issue of how many members a minority class should have before using the seq2seq technique, the minority class proportions for our four datasets vary from approximately 3% to 30% (thanks for the note - we will include such information in Table 1), and our proposed seq2seq technique works well in this range. Since its performance is more stable than other oversampling techniques, we suggest trying this model in the oversampling task as long as the minority class consists of less than 30% of all samples.

---

> > ### Comment · AnonReviewer1 · 2018-12-04
> > **Still unclear what is successfully being learned**
> >
> > You showed that training to learn something about a local neighborhood of the output can help train better neural networks.  I interpret your additional experiments to show that you are learning more than the a trivial single vector pointing to higher-probability outputs, but are not really learning the exact KNN sequence. Similar to Reviewer 2 I still think that a loss function based on the sets of n.n. of the same/different classes is more related to what is actually learnable by this method.  Since you are still not able to provide intuition about what aspect of the class-specific probability distributions is actually being learned, I've downgraded my rating.

---

> > > ### Author Response · Authors · 2018-12-06
> > > **Response 1**
> > >
> > > Thank you for the response. Our intuition is that in classification tasks, the distance between manifolds of different classes should be large and the distance between manifolds of the same class should be small. Therefore, letting neural networks mimic kNN would combine neural networks with the desired distance property of kNN. Our model learns the class label distribution and puts regularizations on it by means of nearest neighbor label distributions & features.
> > >
> > > In such a situation, the order of the nearest neighbors should matter, but should not be very crucial since they are regularizations. We have completed another experiment where we swap the order of the first nearest neighbor and the last nearest neighbor (F-1 score in each dataset is split by “/” below):
> > >
> > > Original V2VSLS: 92.07/94.97/86.24/69.87
> > > Swapped: 91.15/93.81/85.35/68.44
> > >
> > > There is a consistent decrease of F-1 between the original model and the model where we swap the first NN and the last NN. We claim that the order of NN matters, but not drastically. This is due to the loss function having a KL divergence term between the ground truth label and the final predicted label with a high weight, but the weights of regularizations are much smaller so that they have a minor effect.

---

### Meta-Review · Area_Chair1 · 2018-12-14
**nearest neighbour prediction as an auxiliary task**

**Confidence:** 4
**Recommendation:** Reject

**Metareview:**

the proposed approach of predicting k nearest neighbouring examples as an auxiliary task is an interesting idea. however, the submission should have studied further on how those examples are predicted (e.g., sequence prediction is one, but you could try set prediction, or so on) rather than how sequential prediction of nearest neighbours is done together with different types of classifiers (many of which are arguably not necessarily suitable for classification,) which was a sentiment shared by all the reviewers.

more careful investigation of different ways in which nearest neighbour prediction could be incorporated and more careful/thorough analysis on how the incorporation of this auxiliary task changes the behaviours or properties of the representation would make it a much better paper (also with clearer writing.)